# Lysophosphatidylcholines Promote Influenza Virus Reproduction through the MAPK/JNK Pathway in PMA-Differentiated THP-1 Macrophages

**DOI:** 10.3390/ijms25126538

**Published:** 2024-06-13

**Authors:** Min-Ho Cha, Hee-Jeong Choi, Jin-Yeul Ma

**Affiliations:** Korean Medicine (KM) Application Center, Korea Institute of Oriental Medicine (KIOM), Daegu 41062, Republic of Korea; chj1901@kiom.re.kr

**Keywords:** lysophosphatidylcholines, influenza A virus, MAPK pathway, RNA sequencing, THP1 macrophage

## Abstract

Obesity and metabolic syndrome alter serum lipid profiles. They also increase vulnerability to viral infections and worsen the survival rate and symptoms after infection. How serum lipids affect influenza virus proliferation is unclear. Here, we investigated the effects of lysophosphatidylcholines on influenza A virus (IAV) proliferation. IAV particles in the culture medium were titrated using extraction-free quantitative PCR, and viral RNA and protein levels were assessed using real-time PCR and Western blot, respectively. RNA sequencing data were analyzed using PCA and heatmap analysis, and pathway analysis was performed using the KEGG mapper and PathIN tools. Statistical analysis was conducted using SPSS21.0. LPC treatment of THP-1 cells significantly increased IAV proliferation and IAV RNA and protein levels, and saturated LPC was more active in IAV RNA expression than unsaturated LPC was. The functional analysis of genes affected by LPCs showed that the expression of genes involved in IAV signaling, such as suppressor of cytokine signaling 3 (SOCS3), phosphoinositide-3-kinase regulatory subunit 3 (PI3K) and AKT serine/threonine kinase 3 (AKT3), Toll-like receptor 7 (TKR7), and interferon gamma receptor 1 (IFNGR1), was changed by LPC. Altered influenza A pathways were linked with MAPK and PI3K/AKT signaling. Treatment with inhibitors of MAPK or PI3K attenuated viral gene expression changes induced by LPCs. The present study shows that LPCs stimulated virus reproduction by modifying the cellular environment to one in which viruses proliferated better. This was mediated by the MAPK, JNK, and PI3K/AKT pathways. Further animal studies are needed to confirm the link between LPCs from serum or the respiratory system and IAV proliferation.

## 1. Introduction

Obesity and metabolic syndrome contribute greatly to the development of various diseases such as atherosclerosis, cardiovascular diseases, and type 2 diabetes [1,2,3,4]. Additionally, obesity increases vulnerability to viral infections, decreases survival after infection, and is associated with worth outcomes after infection [5,6,7,8]. During the recent pandemic, the mortality rate due to coronavirus disease 2019 was higher in patients with obesity than in people without [7]. Metabolic complications, nutritional deficiency, physical inactivity, and chronic hormonal imbalance are known causes of the higher rate of viral infections due to obesity [5]. However, the molecular mechanisms involved in the higher susceptibility and longer recovery after infection observed in patients with obesity remain unclear.

Obesity causes changes in various hematological indicators, such as the levels of immune cells, lipids, and cytokines in the blood [9]. In particular, serum lipids such as triglycerides, cholesterol, and ceramides are significantly changed in patients with obesity [10,11]. The development of lipidomic analysis has revealed that different types of lipids are differently related to obesity [12]. Among serum lipids, the levels of lysophosphatidylcholines (LPCs), which function as signaling molecules derived from phosphatidylcholines by phospholipase A2, have been associated with obesity [13,14].

The human influenza virus is a member of the Orthomyxoviridae family. Depending on its genome structure, it is classified into types A, B, C, and D [15]. Human influenza A viruses (IAVs) are the only type of influenza virus causing flu pandemics [16]. Since the gene structures of influenza A viruses frequently evolve due to antigenic drifts or antigenic shifts [17,18], the symptoms and severity of these viral infections also vary depending on individual differences.

Replication of IAVs occurs through several steps, including adsorption, RNA synthesis, morphogenesis (including viral release), and various other mechanisms occurring in the host cell [19,20]. Recent studies have shown that cellular lipid content and protein–lipid interactions influence IAV replication [21,22].

In the present study, we investigated the effects of LPCs on IAV proliferation in THP-1 macrophages as the cellular state of macrophages is modified by LPCs.

## 2. Results

### 2.1. LPCs Decrease the Survival of Cells Infected with IAV

To determine the effects of LPCs on virus proliferation, we first assessed cell viability after IAV infection. Cell viability examined 24 and 36 h after IAV infection was significantly reduced in the 20 μM and 40 μM LPC-pretreated group compared with the IAV group (*p* < 0.01) (Figure 1A). The amount of IAV secreted into the supernatant was also increased by LPCs (Figure 1B,C). To confirm the changes in virus infectivity caused by LPCs, changes in the levels of intracellular viral genomic DNA were determined 0 h after virus infection. Although the levels of viral genome were slightly increased, there was no significant difference (Figure 1D). Altogether, these data suggest that pretreatment with LPCs affected virus proliferation.

### 2.2. LPCs Increase Viral RNA and Protein Expression

To investigate whether LPC pretreatment increased viral RNA expression, we determined the level of nine viral RNAs (NS1, NP, M1, M2, NA, HA, PA, PB1, and PB2) by RT-qPCR. Compared to the IAV group, the levels of all viral RNAs were significantly, more than threefold, increased by the pretreatment with LPCs at a concentration of 40 μM (Figure 2A). Then, we detected viral proteins from the cells pretreated with low (10 μM) and high (40 μM) amounts of LPCs using Western blot. Consistent with the viral RNA levels, the expression of viral proteins (PA1, NP, NS1, M1, PB1, and PB2) was significantly increased by the high dose of LPCs (Figure 2B). These data show that LPCs increase the expression level of viral RNA as well as protein in THP-1 cells.

### 2.3. Saturated LPCs Further Increase IAV RNA Levels

LPCs are divided into saturated and unsaturated types depending on whether the acyl group of the LPCs contains a double bond. Since saturated LPCs are more active in THP-1 cells [23], we determined which type of LPCs impacted IAV proliferation the most. Figure 3 shows that LPC16:0 and LPC18:0 increased viral gene (NS1, NP, M1, M2, NA, HA, PA, PB1, and PB2) expression by more than threefold compared to the group treated with IAV only. LPC18:1, which contains one double bond on the acyl group, also increased the expression of virus genes, but to a lesser extent than LPC16:0 or LPC18:0. Therefore, the result implies that saturated types of LPCs lead to more significant increases in viral genes than unsaturated types of LPCs.

### 2.4. LPCs Alter Host Gene Expression

To replicate their genome and exit cells, viruses alter the expression of the host genes to change the host environment. We analyzed RNA profiling changes to infer LPC-induced changes in the host cell environment and determine how LPCs altered the cell states to stimulate the proliferation of the virus. Gene expression profiles in THP-1 macrophages were significantly impacted by LPCs (Figure 4A), and the mRNA levels of 1182 genes were significantly different by more than twofold (decreased levels of 606 genes and increased expression of 582 genes) in LPC-treated cells (Figure 4B).

Pathway analysis using the Kyoto Encyclopedia of Genes and Genomes (KEGG) database showed that genes involved in the mitogen-activated protein kinase (MAPK) and peroxisome proliferator-activated receptor (PPAR) signaling pathways were significantly affected by LPCs. The KEGG pathways affected by LPCs are listed in Appendix A.

Among the pathways affected by LPCs, various virus infection pathways, such as human papillomavirus infection and Kaposi sarcoma-associated herpes virus infection, and viral protein interaction cytokines and cytokine receptors were also included. The data indicate that LPCs might change the host cell environment into one favorable for virus propagation.

### 2.5. LPCs Affect Influenza A Pathway and Their Related Pathways

We selected the genes involved in influenza A pathways from the KEGG database and checked their expression using RNA expression data (Figure 5A). Genes activated by the NS1 viral protein, such as suppressor of cytokine signaling 3 (SOCS3) and phosphoinositide 3-kinase (PI3K), were highly expressed in LPC-treated cells. On the other hand, the levels of mRNAs encoding interferon gamma receptor (IFNGR) and Toll-like receptor 7 (TLR7), which are involved in defense against IAV proliferation, were decreased.

Influenza A pathways are related with other pathways such as the MAP kinase signaling pathway, RIG signaling pathway, JNK signaling pathway, etc. (Figure 5A). Pathway network analysis was performed using the PathIN database to identify the pathways linked with influenza A among those affected by LPCs. The MAPK signaling, PI3K/AKT signaling, apoptosis, and terpenoid synthesis pathways were highly linked with influenza A pathways (Figure 5B). In particular, the levels of pp38 and pJNK, key molecules in MAPK signaling, were highly increased by LPC, but the pAKT level did not change (Figure 5C). These results suggest that LPCs affected pathways involved in IAV propagation, providing an environment that promoted IAV proliferation.

### 2.6. MAPK and PI3K Inhibitors Decreased Viral RNA Levels

Since pp38 and pJNK, key proteins in the MAPK and JNK signaling pathways, were increased by LPCs, in order to confirm that the increase in viral proliferation by LPCs was mediated by the MAPK and JNK signaling pathways, viral RNA expression was analyzed in cells co-treated with LPCs and SB203580 (MAPK inhibitor) or SP600125 (JNK inhibitor). (Figure 6). MAPK inhibitor treatment significantly attenuated the increase in M2, NA, HA PA, PB1, and PB2 viral RNAs’ expression induced by LPCs. The JNK inhibitor and PI3K inhibitor (LY294002) decreased viral PB1 and PB2 RNA levels, which were increased by LPCs. These results imply that the increase in viral proliferation induced by LPCs was partially mediated by the MAPK and PI3K/AKT pathways.

## 3. Discussion

Influenza is a representative infectious disease that is transmitted through the air and causes mild respiratory symptoms, such as coughing and a runny nose. It can also be a serious disease that can lead to death in severe cases. According to a report from the World Health Organization (WHO), at least 5–10% of the world’s population is infected with influenza each year, resulting in 290,000 to 650,000 deaths [23]. People who are vulnerable to influenza are pregnant women, children under 5 years of age, elderly people over 50 years of age, and patients with respiratory diseases, diabetes, asthma, and cardiovascular diseases.

Obesity is also a vulnerability factor for infection with influenza. A meta-analysis performed by Zhao et al. showed that the severity of and mortality resulting from influenza in patients with obesity was more than 1.5- and 1.99-fold higher, respectively, compared with those in subjects without obesity [24]. Neidich et al. reported that vaccinated patients with obesity have a twofold higher risk of contracting influenza or influenza-like illness than vaccinated subjects without obesity because of an impaired cellular immune response against influenza virus [25]. Moreover, the production of antibodies against influenza is decreased in patients with diet-induced obesity [26].

Lipidomic analyses in patients with obesity have shown that serum LPC levels are linked to obesity indices, but these data are controversial [14,16]. LPCs are derived from phosphatidylcholine by phospholipase A2. They are minor phospholipids found in cells and serum and are quickly metabolized to phosphatidylcholine or lysophosphatidic acid. LPCs are associated with various diseases, such as atherosclerosis and allergic airway disease manifestation [27,28]. In subjects with asthma, the levels of saturated LPCs (LPC16:0 and LPC18:0) are elevated in broncho-alveolar lavage fluids [29]. However, which factors increase virus infection and proliferation in patients with obesity remain unclear. 

In the present study, we showed that IAV proliferation increased in THP-1 cells exposed to high concentrations of LPCs. (Figure 1), and expression of viral RNAs and proteins was also higher (Figure 2B). A previous study reported by Cha et al. showed that saturated LPCs (LPC16:0 and LPC18:0) significantly increased the cholesterol synthesis pathway, but unsaturated LPCs (LPC18:0) did not [30]. For this reason, we compared viral RNA expression in the presence of different types of LPCs according to the degree of saturation of acyl groups. The proliferation of IAV was increased with all LPCs, but it proliferated more in THP-1 cells pretreated with saturated LPCs than in cells exposed to unsaturated LPCs (Figure 3).

Viral proliferation depends on host cell conditions, as viruses use the host cell machinery to replicate their genome and package it into viral proteins. For example, NS1 is known to inhibit apoptosis by interacting with the PI3K/AKT pathway, thereby securing sufficient time for virus propagation [31]. PB1-F2 binds to the nucleotide-binding oligomerization domain-like receptor (NLRX1) and prevents mitochondrial apoptosis [32]. Therefore, the host condition affects the propagation of the virus, as it might change the state of proteins’ implication in virus replication and apoptosis.

RNA profiling analysis was employed to confirm the intracellular environment change caused by LPCs and revealed that the expression of SOCS3, PI3K, and protein kinase B (PKB) was increased, while the expression of INFGR, interleukin-1b, and NLRX1 was suppressed (Figure 5A). SOCS3 is known to inhibit cytokine signaling by blocking the JAK/STAT pathway [33]. PI3K and NLRX1, which are involved in apoptosis, were also differently expressed upon LPC treatment [31,34]. This indicated that the increased SOCS3 and PI3K and decreased NLRX1 expressions induced by LPCs may promote IAV proliferation by inhibiting the cytokine-mediated virus defense system and increasing cell survival.

Signaling pathways interact, and influenza A signaling was linked with JAK/STAT and MAPK signaling. The network analysis of pathways affected by LPCs showed that influenza A signaling pathways were associated with apoptosis, MAPK signaling, T cell receptor signaling, and cell cycles (Figure 5B).

Jing et al. reported that LPCs activate p38 and p42/44 MAPK in THP-1 cells [35], and Tan et al. also showed that LPCs activate p38, which is a host protein essential for virus infection and replication [36]. Inhibition of MAPK signaling by SB203580, a p38 inhibitor, suppresses IAV replication by inhibiting virus-induced MAPK activation [35]. Our findings suggest that p38 was activated by LPCs and promoted viral proliferation after infection. Co-treatment with LPCs and SB203580 before infection attenuated IAV gene expression in THP-1 macrophages compared with cells pretreated with LPCs (Figure 6). LPCs also stimulated the PI3K/AKT pathway by increasing PI3K expression, indicating that LPCs might suppress virus-induced apoptosis and increase IAV replication. Co-treatment with PI3K inhibitors and LPCs before infection attenuated viral gene expression (Figure 6).

The present study is the first to demonstrate that LPCs promote viral replication. Our findings provide an insight into the molecular mechanisms involved in the greater vulnerability of patients with various diseases, especially metabolic diseases, to viral infections. Additional research in animals is needed to further identify the mechanisms of viral infection vulnerability.

## 4. Materials and Methods

### 4.1. Reagents

LPCs (LPC16:0, LPC18:0, and LPC18:1) were purchased from Sigma-Aldrich (St Louis, MO, USA), reconstituted in dimethyl sulfoxide or methanol at a concentration 1000 times the final concentration used for cell incubation, and stored at −20 °C. Phorbol 12-myristate 13-acetate (PMA) and SB203580 were also purchased from Sigma-Aldrich. LY294002 and SP600125 were obtained from Cell Signaling Technology (Danvers, MA, USA). Antibodies against the influenza virus proteins polymerase acidic (PA), nucleoprotein (NP), non-structural protein 1 (NS1), matrix-1 (M1), polymerase basic 1 (PB1), and PB2 were obtained from GeneTex, Inc. (Irvine, CA, USA), and antibodies for JNK, p-JNK, AKT, and p-AKT were purchased from Cell Signaling Technology (Danvers, MA, USA). Antibodies for phospho-p38, p38, and β-actin were purchased from Santa Cruz Biotechnology (Dallas, TX, USA).

### 4.2. Cell Culture and Virus Infection

THP-1 cells (TIB-202™) were purchased from ATCC (Manassas, VA, USA) and maintained in RPMI (Roswell Park Memorial Institute) medium (Hyclone, Logan, UT, USA) containing 10% fetal bovine serum and 100 U/mL of penicillin and streptomycin at 37 °C with 5% CO_2_.

IAV H1N1 A/PR8/34(H1N1) was obtained from Dr. Jong-Soo Lee (Chungnam National University, Daejeon, Republic of Korea) and was amplified in allantoic fluid of 10-day-old chicken embryos. The virus was titrated to 109/mL and stored at −70 °C until use.

THP-1 cells were derived to macrophages using 200 ng/mL PMA. Treatments with LPCs or other chemicals were performed according to Cha et al. (2019) [30]. After treatment with LPCs for 8 h, cells were infected with 105/mL IAV for 2 h. Then, the medium was replaced with fresh medium.

### 4.3. Cytotoxicity Test

The cell viability of THP-1 macrophages after IAV infection was detected using CCK-8 reagent (Dojindo, Rockville, MD, USA) following the manufacturer’s protocol. THP-1 macrophages seeded in a 96-well plate (1 × 105 cells/well) were incubated with LPCs for 8 h and infected with IAV for 24 or 36 h. After adding the CCK-8 solution, the cells were incubated at 37 °C for 2 h, and the absorbance was measured at 450 nm using a microplate reader (Promega, Madison, WI, USA).

### 4.4. Virus Titration from Cell Supernatant

The amount of virus excreted into the medium was detected using the extraction-free reverse transcription–polymerase chain reaction (RT-PCR) method described by Smyrlaki et al. [37]. Briefly, cell supernatants containing virus were incubated at 95 °C for 10 min to inactivate the virus. Then, the supernatants were subjected to reverse transcription using the iScript RT kit (Bio-Rad, Hercules, CA, USA), and the levels of viral RNA were quantified using the Bio-Rad Real-Time PCR system (Bio-Rad, Hercules, CA, USA) and SsoAdvanced™ Universal SYBR^®^ Green Supermix (Bio-Rad, Hercules, CA, USA). Primers recognizing the M2 and NP viral genomic regions were used for quantification.

### 4.5. Quantitative PCR (qPCR)

After IAV infection, viral RNA from THP-1 macrophages pretreated by LPCs in the presence or absence of other chemicals was detected using RT-qPCR. Briefly, RNA from cells was extracted using the easy-spin™ Total RNA Extraction Kit (iNtrON, Seoul, Republic of Korea) following the manufacturer’s protocol. After RT using the iScript RT kit (Bio-Rad, Hercules, CA, USA), the levels of viral RNA were measured using the Bio-Rad Real-Time PCR system (Bio-Rad, Hercules, CA, USA) and SsoAdvanced™ Universal SYBR^®^ Green Supermix (Bio-Rad, Hercules, CA, USA). The primers used for qPCR are listed in Appendix A.

### 4.6. Western Blot

Viral proteins from THP-1 macrophages were detected after LPC treatment and IAV infection using Western blot. Briefly, cells were lysed in radioimmunoprecipitation assay buffer. Proteins were separated by sodium dodecyl sulfate-polyacrylamide gel electrophoresis (12%) and transferred to polyvinylidene fluoride membranes (Millipore) for 2 h. After blocking the membranes with 5% bovine serum albumin (GenDEPOT, Katy, TX, USA) for 1 h, they were washed with Tris-buffered saline including Tween-20 (TBST) three times for 10 min. Then, the membranes were incubated with specific primary antibodies overnight at 4 °C. Afterwards, the membranes were washed thrice with TBST and incubated with horseradish peroxidase-conjugated secondary antibodies for 1 h at room temperature. After three washes, the expression of proteins was detected using the Pierce™ ECL Western Blotting Substrate (Thermo Fisher Scientific, Waltham, MA, USA).

### 4.7. Prediction of Putative Transcription Factor Binding Sites

To detect potential transcription factor binding sites (TFBSs) that may be affected by methylation at the investigated CpG sites in the *NPR2* promoter region, a search on the PROMO database was performed [29]. Predictions were made after limiting the search to the human species and setting the sequence dissimilarity threshold within 10%.

### 4.8. RNA Profile Analysis

RNA sequencing data were taken from a previous study conducted by Cha et al. [30]. A principal component analysis (PCA) and a heatmap analysis of genes differently expressed in the presence of LPCs were performed using ClustVis (https://biit.cs.ut.ee/clustvis; accessed on 2 November 2023), a webtool for visualizing the clustering of multivariate data. Kyoto Encyclopedia of Genes and Genomes (KEGG) functional annotation of genes differently expressed in the presence of LPCs was conducted using the KEGG mapper (https://www.genome.jp/kegg/mapper: accessed on 6 November 2023) and ShinyGO (http://bioinformatics.sdstate.edu: accessed on 6 November 2023) with a restricted false discovery rate (FDR) < 0.05. Pathway networks linked with influenza A pathways were analyzed using PathIN tools (https://pathin.cing-big.hpcf.cyi.ac.cy: accessed on 30 November 2023) [38].

### 4.9. Statistical Analysis

Data are expressed as means ± standard deviations (SDs), and statistical analysis was performed using a two-tailed Student t-test using SPSS21.0 (Korea) and statistical significance. A *p*-value < 0.05 indicated statistical significance.

## 5. Conclusions

Here, we showed that LPCs stimulated IAV proliferation, possibly by activating MAPK and PI3K/AKT signaling in THP-1 macrophages. Although further studies are needed to determine how LPCs increase virus proliferation, the present report revealed a link between changes in the lipid profile and increased viral infection rates.

## Figures and Tables

**Figure 1 ijms-25-06538-f001:**
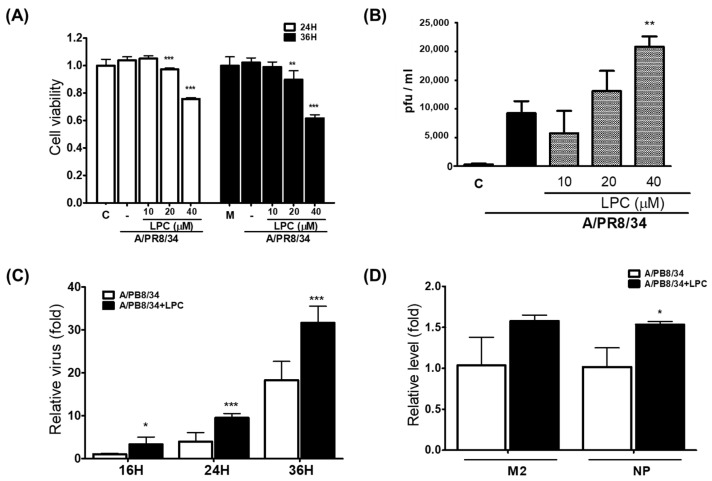
Increase in IAV titer after LPC treatment. (**A**) Viability of THP-1 macrophages pretreated with different doses of LPCs and exposed to IAV for 24 or 36 h. (**B**) Number of virus particles released into the culture medium of cells treated with LPCs before IAV infection. Color boxs indicated control (□), IAV (■) and IAV with LPC (▒). (**C**) Relative quantification of the number of virus particles released into the culture medium at different times after IAV infection of cells pretreated with LPCs for 8 h. (**D**) Relative amount of viral genome in cells immediately after viral infection. THP-1 macrophages were treated with LPCs at the indicated concentrations for 8 h and then infected with IAV for 2 h. After replacing the medium with fresh medium, cells were incubated for the indicated times. Cell viability was determined using the CCK-8 assay, and IAV titer was detected using RT-qPCR. The data are representative of three independent experiments and are presented as means ± SDs of three independent experiments. Statistical significance was assessed using an unpaired Student *t*-test. *** *p* < 0.001, ** *p* < 0.005, * *p* < 0.05 compared with the values obtained for cells infected with IAV.

**Figure 2 ijms-25-06538-f002:**
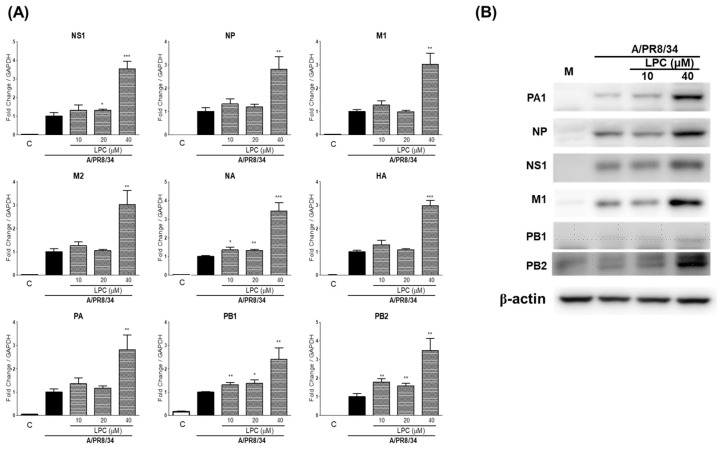
Increase in viral RNA and protein levels by LPCs. (**A**) Relative expression levels of IAV genes in THP-1 cells treated with different dosages of LPCs. Color boxes indicated control (□), IAV (■) and IAV with LPC (▒). (**B**) The protein level of IAV proteins in THP-1 cells treated with LPCs. RNA levels were determined using RT-qPCR, and protein expression was detected using Western blotting. Data represent the means ± SDs of three independent experiments. Statistical significance was assessed using an unpaired Student *t*-test. *** *p* < 0.001, ** *p* < 0.005, * *p* < 0.05 compared with the values obtained for cells infected with IAV. M2: matrix-2; M1: matrix-1; PA: polymerase acidic protein; PB1: polymerase basic 1; NP: nucleoprotein; NS1: non-structural protein; PB2: polymerase basic 2; HA: hemagglutinin; NA: neuraminidase.

**Figure 3 ijms-25-06538-f003:**
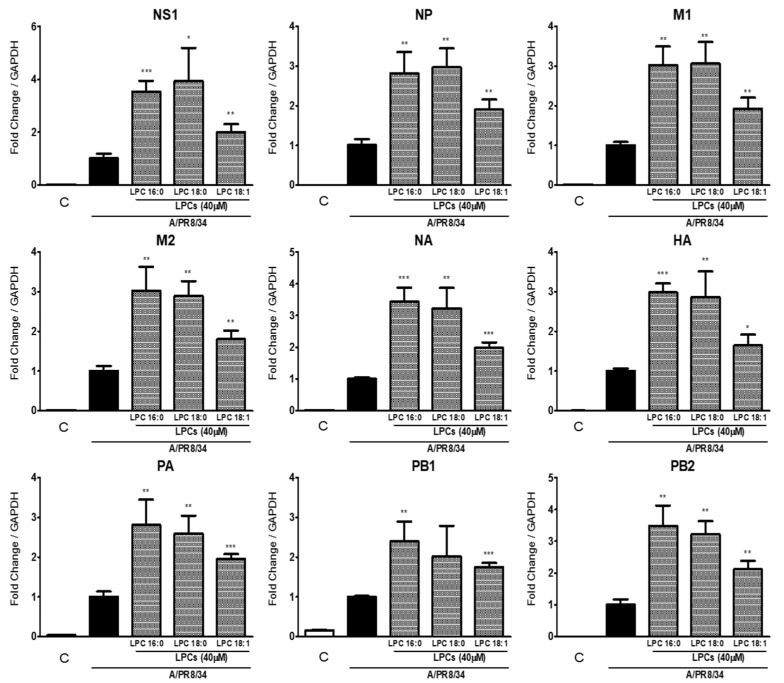
Differential expression of viral RNAs according to the type of LPCs. The relative expression levels of IAV genes were determined using RT-qPCR. Data represent the means ± SDs of three independent experiments. Statistical significance was assessed using an unpaired Student *t*-test. *** *p* < 0.001, ** *p* < 0.005, * *p* < 0.05 compared with the values obtained for cells infected with IAV. Color boxes indicated control (□), IAV (■) and IAV with LPCs (▒). M2: matrix-2; M1: matrix-1; PA: polymerase acidic protein; PB1: polymerase basic 1; NP: nucleoprotein; NS1: non-structural protein; PB2: polymerase basic 2; HA: hemagglutinin; NA: neuraminidase.

**Figure 4 ijms-25-06538-f004:**
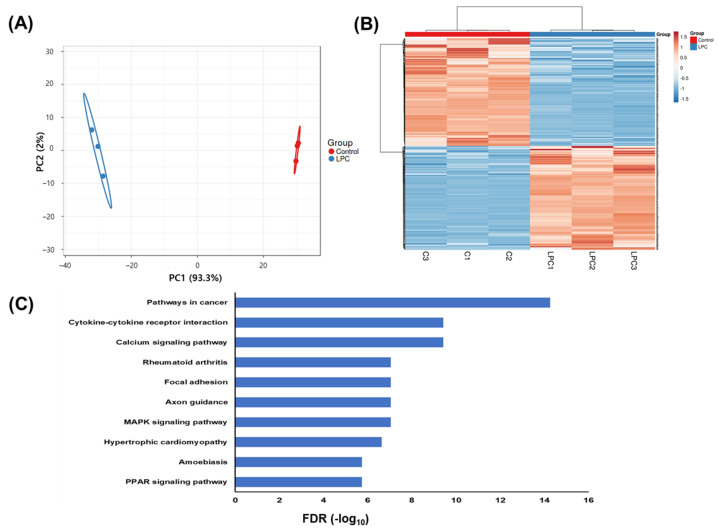
RNA expression profiles in THP-1 cells treated with LPCs. (**A**) PCA of RNA profiles in control cells and cells treated with 40 µM LPCs. (**B**) Heatmap of genes with a more than twofold expression changes in response to LPCs (*p* < 0.01). Heatmap was obtained using ClustVis (https://biit.cs.ut.ee/clustvis/; accessed on 2 November 2023). (**C**) KEGG pathway analysis of 1082 genes with decreased or enhanced expression in response to LPCs.

**Figure 5 ijms-25-06538-f005:**
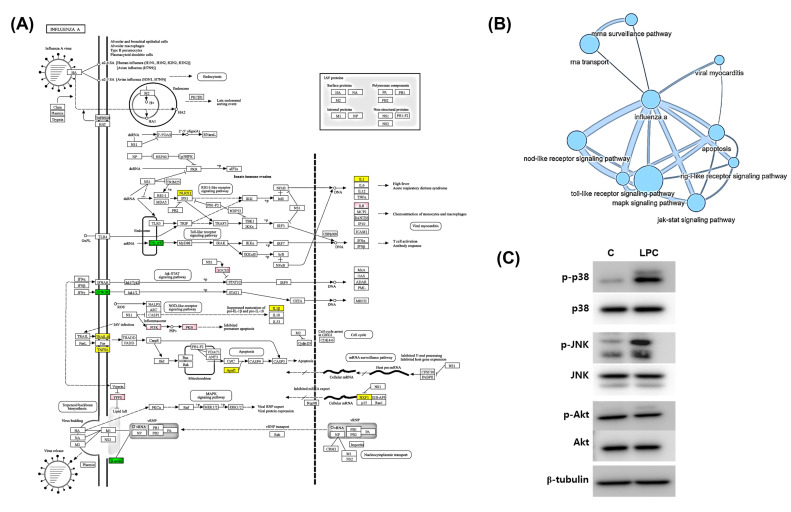
Effects of LPCs on influenza A signaling pathways. (**A**) Genes involved in influenza A signaling pathways and altered by LPC treatment of THP-1 cells. Pink indicates genes of which the expression increased 2-fold upon LPC treatment, respectively. Green and yellow indicate genes with a more than 2- or 1.5-fold decrease in expression upon LPC treatment, respectively. (**B**) Interaction of pathways linked with influenza A signaling and affected by LPCs. (**C**) The protein level of p38, JNK, and AKT and their phosphorylated forms in THP-1 cells treated with LPCs. The protein expression was detected using Western blotting. p38: mitogen-activated protein kinase P38 alpha; JNK: C-Jun N-terminal kinase 1; AKT: AKT serine/threonine kinase 1.

**Figure 6 ijms-25-06538-f006:**
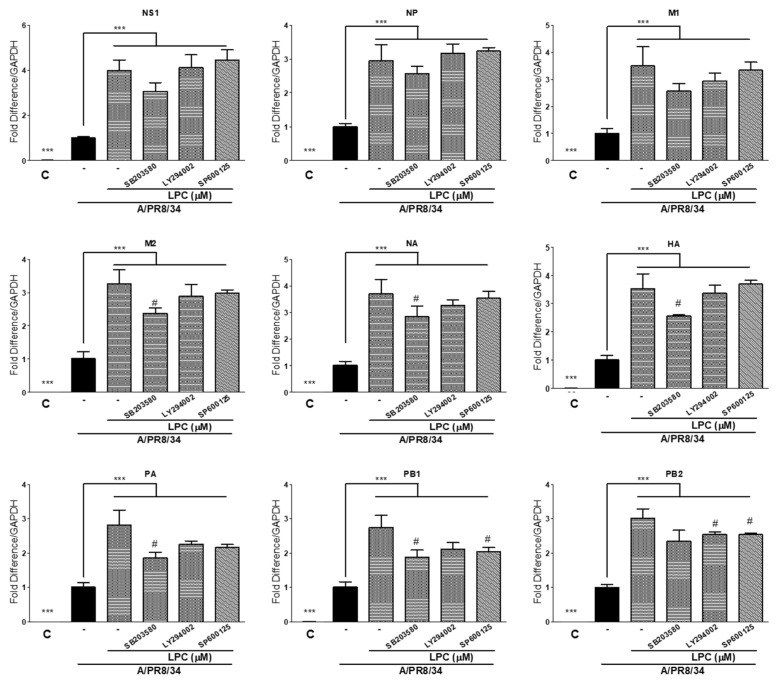
Attenuation of viral gene expression by MAP kinase inhibitor (SB203580), JNK inhibitor, and PI3K inhibitor (LY294002) in LPC-treated THP-1 cells. Relative expression levels of IAV genes were determined using RT-qPCR. Data represent the means ± SDs of three independent experiments. Statistical significance was assessed using an unpaired Student *t*-test. *** *p* < 0.001 compared with values obtained for cells infected with IAV, # *p* < 0.05 compared with values obtained for cells infected with IAV and treated with LPC. Color boxes indicated control (□), IAV (■) and IAV with LPC and inhibitors (▒). M2: matrix-2; M1: matrix-1; PA: polymerase acidic protein; PB1: polymerase basic 1; NP: nucleoprotein; NS1: non-structural protein; PB2: polymerase basic 2; HA: hemagglutinin; NA: neuraminidase.

## Data Availability

All datasets used and analyzed during the current study are available from the corresponding author on reasonable request.

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
