# Peer review of "Lysophosphatidylcholines Promote Influenza Virus Reproduction through the MAPK/JNK Pathway in PMA-Differentiated THP-1 Macrophages"

_ijms, 2024, doi:10.3390/ijms25126538_

Round 1

Reviewer 1 Report

Comments and Suggestions for Authors

Dear editor,

In this manuscript, it is demonstrated that LPCs can promote viral replication by investigating the effects of lysophosphatidylcholines on influenza A virus (IAV) proliferation. However, many comments are still needed to be addressed below:

Major comments:

1.     In the “Abstract” section, it was stated that “Functional analysis of genes affected by LPCs showed that expression of genes involved in influenza A signaling was changed.” This description is too broad and vague, please rewrite the description.

2.     In section 2.3 “Saturated LPCs increase IAV RNA levels”, please give more description about the results and discuss the possible reasons why saturated LPCs could increase the IAV RNA levels.

3.     The results of Figure 5C was not cited in the main text.

4.     What is the deference between the nine figures in Figure 6? Please give more information in figure legend and the main text.

Comments on the Quality of English Language

Minor editing of English language is required.

Author Response

Major comments:

  1. In the “Abstract” section, it was stated that “Functional analysis of genes affected by LPCs showed that expression of genes involved in influenza A signaling was changed.” This description is too broad and vague, please rewrite the description.

 -> Answer : --> Add genes involved in the IAV signaling pathway affected by LPC in abstract

.

Functional analysis of genes affected by LPCs showed that expression of genes involving in IAV signaling, such as Suppressor Of Cytokine Signaling 3 (SOCS3), Phosphoinositide-3-Kinase Regulatory Subunit 3 (PI3K) and AKT Serine/Threonine Kinase 3 (AKT3), toll Like Receptor 7 (TKR7) and interferon Gamma Receptor 1 (IFNGR1), were changed by LPC.

line 19-23

  1. In section 2.3 “Saturated LPCs increase IAV RNA levels”, please give more description about the results and discuss the possible reasons why saturated LPCs could increase the IAV RNA levels.

 -> Answer : --> corrects the description in result section 2.3 and add discussion

2.3. Saturated LPCs further increase IAV RNA levels

LPCs are divided into saturated and unsaturated types depending on whether the acyl group of LPCs contains a double bond. Since saturated LPCs are more active in THP-1 cells [23], we determined which type of LPCs impacted IAV proliferation the most. Fig. 3 showed that LPC16:0 and LPC18:0 increased viral gene (NS1, NP, M1, M2, NA, HA, PA, PB1, PB2) expression by more than three folds compared with only IAV treated group. LPC18:1, which contain one double bond on the acyl group, also increased the expression of virus genes, but to a lesser extent than LPC16:0 or LPC18:0. Therefore, the result implied that saturated types of LPCs showed more significant increased viral genes than unsaturated types of LPCs

line 112-120 in result

IAV proliferation increased in THP-1 cells exposed to high concentrations of LPCs (Fig. 1), and expression of viral RNAs and proteins was also higher (Fig. 2B). Previous study reported by Cha et al. showed that saturated LPCs (LPC16:0 and LPC18:0) was significantly increased cholesterol synthesis pathway, but unsaturated LPC (LPC18:0) was not the pathway [23]. For this reason, we compared viral RNA expression in presence of different types of LPCs according to the degree of saturation in acyl group. The proliferation of IAV was increased with all LPCs, but IAVs proliferated more in THP-1 cells pretreated with saturated LPCs than in cells exposed to unsaturated LPCs (Fig. 3).

line 232-239 in discussion

  1. The results of Figure 5C was not cited in the main text.

 -> Answer : --> Add the cite of Fig. 5C

line ; 168

  1. What is the deference between the nine figures in Figure 6? Please give more information in figure legend and the main text.

 -> Answer : --> Describe about Fig. 6 at text in detail.

Since pp38 and pJNK, key proteins in the MAPK and JNK signaling pathways, were increased by LPC, in order to confirm that the increase in viral proliferation by LPC was mediated by the MAPK and JNK signaling pathways, viral RNA expression was analyzed in cells co-treated with LPC and SB203580 (MAPK inhibitor) or SP600125 (JNK inhibitor). (Fig. 6). MAPK inhibitor treatment significantly attenuated the increase in M2, NA, HA PA, PB1 and PB2 viral RNAs expression induced by LPCs. The JNK inhibitor and PI3K inhibitor (LY294002) decreased viral PB1 and PB2 RNAs level increased by LPC. These results implied that the increase of viral proliferation induced by LPCs was partially mediated by MAPK and PI3K/AKT pathways.

line : 184-192

Reviewer 2 Report

Comments and Suggestions for Authors

The manuscript by Cha M-H et al entitled “Lysophosphatidylcholines promote influenza virus reproduction through MAP/JNK pathway in PMA-differentiated THP-1 macrophage” is a study around the effect of lysophosphatidylcholines on influenza proliferation.

Few minor modifications are required before the publication:

- introductory part: page 2, line 53. The sentence around the vaccine could be misunderstood and should be reformulated. Indeed, although to date, there is no a universal vaccine to protect the population from the influenza infection, there is an annual tetravalent vaccine (included of 1 or 2 influenza A strains) that protects from annual epidemics.

- results part: in general, a more detailed description of the assays and, above all, of the obtained results is strictly necessary. Some of the chapter are very short (eg 2.2 or 2.3) and, in this way, some essential information could be lost.

- page 2, lines 62-63: from the figure 1A, it seems that the cell viability was not significantly reduced in the LPC-pretreated group, as instead reported in the text. Please, check, and eventually correct.

-Figure 6: in the figure description please report that the acronym of the JNK inhibitor in brackets, as reported for the other 2 compounds.

Author Response

Few minor modifications are required before the publication:

Comment 1

- introductory part: page 2, line 53. The sentence around the vaccine could be misunderstood and should be reformulated. Indeed, although to date, there is no a universal vaccine to protect the population from the influenza infection, there is an annual tetravalent vaccine (included of 1 or 2 influenza A strains) that protects from annual epidemics.

-> Answer : --> The reviewer's point is correct. This study is about how blood lipid composition varies from person to person, and in particular, the difference in blood LPC concentration in obese people affects the increase in viruses. Therefore, the pointed out sentence was modified as follows.

Because the gene structures of influenza A viruses frequently evolve due to antigenic drifts or antigenic shifts [17,18], symptoms and severity by viral infection also vary depending on individual differences.

line 53-56

Comment 2

- results part: in general, a more detailed description of the assays and, above all, of the obtained results is strictly necessary. Some of the chapter are very short (eg 2.2 or 2.3) and, in this way, some essential information could be lost.

-> Answer : --> Correct the description of results as below

2.2. LPCs increase viral RNA and protein expression

To investigate whether LPC pretreatment increased viral RNA expression, we determined the level of nine viral RNAs (NS1, NP, M1, M2, NA, HA, PA, PB1, PB2) by RT q-PCR. Compared to IAV group, the levels of all viral RNAs were significantly, more than three folds, increased by the pretreatment with LPCs at a concentration of 40 μM (Fig. 2A). Then, we detected viral proteins from the cells pretreated with low (10 μM ) and high (40 μM) of LPC using western blot. Consistent with viral RNA levels, the expression of viral proteins (PA1, NP, NS1, M1, PB1, PB2) was significantly increased by a high dose of LPCs (Fig.2B). These data showed that LPCs induce the expression level of viral RNA as well as protein in THP-1 cells.

line : 89-98

2.3

LPCs are divided into saturated and unsaturated types depending on whether the acyl group of LPCs contains a double bond. Since saturated LPCs are more active in THP-1 cells [23], we determined which type of LPCs impacted IAV proliferation the most. Fig. 3 showed that LPC16:0 and LPC18:0 increased viral gene (NS1, NP, M1, M2, NA, HA, PA, PB1, PB2) expression by more than three folds compared with only IAV treated group. LPC18:1, which contain one double bond on the acyl group, also increased the expression of virus genes, but to a lesser extent than LPC16:0 or LPC18:0. Therefore, the result implied that saturated types of LPCs showed more significant increased viral genes than unsaturated types of LPCs.

line : 112-120

2.6

Since pp38 and pJNK, key proteins in the MAPK and JNK signaling pathways, were increased by LPC, in order to confirm that the increase in viral proliferation by LPC was mediated by the MAPK and JNK signaling pathways, viral RNA expression was analyzed in cells co-treated with LPC and SB203580 (MAPK inhibitor) or SP600125 (JNK inhibitor). (Fig. 6). MAPK inhibitor treatment significantly attenuated the increase in M2, NA, HA PA, PB1 and PB2 viral RNAs expression induced by LPCs. The JNK inhibitor and PI3K inhibitor (LY294002) decreased viral PB1 and PB2 RNAs level increased by LPC. These results implied that the increase of viral proliferation induced by LPCs was partially mediated by MAPK and PI3K/AKT pathways.

line : 184-192

Comment 3

- page 2, lines 62-63: from the figure 1A, it seems that the cell viability was not significantly reduced in the LPC-pretreated group, as instead reported in the text. Please, check, and eventually correct.

-> Answer : --> Cell viability was decreased at 20 and 40uM of LPC. Therefore, explanation for thes is clearly stated as below.

Cell viability examined 24 and 36 h after IAV infection was significantly reduced in the 20μM and 40μM LPC-pretreated group compared with IAV group (p<0.01) (Fig. 1A).

line : 66-68.

Comment 4

-Figure 6: in the figure description please report that the acronym of the JNK inhibitor in brackets, as reported for the other 2 compounds.

-> Answer : --> add the acronym of the JNK inhibitor

line : 190

Round 2

Reviewer 1 Report

Comments and Suggestions for Authors

I carefully read the paper and I could see that the authors corrected the paper as suggested by the reviewers. I found that this paper can now be published in its present version.